# Designing Culturally Responsive Education Strategies to Cultivate Young Children’s Cultural Identities: A Case Study of the Development of a Preschool Local Culture Curriculum

**DOI:** 10.3390/children9121789

**Published:** 2022-11-22

**Authors:** Yi-Huang Shih

**Affiliations:** Department of Early Childhood Education and Care, Minghsin University of Science and Technology, Hsinchu 30401, Taiwan; shih262@gmail.com; Tel.: +886-03-0915306690

**Keywords:** culturally responsive education, cultural identity, local culture curriculum, preschool

## Abstract

The researcher investigated how the teachers at the preschool integrated the vision of the preschool, educational goals, and community resources to develop and implement the local culture curriculum as well as what problems they encountered in the process. Thereafter, the researcher developed strategies that can be used to solve such problems. The researcher discovered that the most important aspects of the local culture curriculum were (1) its ability to pique the children’s interest in history and the humanities through the lens of the children’s experiences at the market, and (2) its integration of local culture and the community, specifically through interactions between young children and adults at a vegetable market. The implementation of the local culture curriculum involved the following steps: (1) establishing a thematic network; (2) planning the activities, including exploring the children’s previous experiences and adjusting the curriculum and teaching methods accordingly; (3) conducting the activities, including visiting and participating in work at the market; (4) evaluating the activities; and (5) creating thank you cards and formulating plans for store renovation. Some of the obstacles the teachers encountered in the development and implementation of the vegetable-market-themed local culture curriculum were (1) shortages of preschool teachers willing to teach outdoor courses, (2) concerns about young children’s safety, and (3) young children’s lack of experience. Finally, on the basis of my conclusion, the researcher developed the following culturally responsive education strategies: (1) integrating aspects of local culture and the community into courses, (2) situating local culture courses in learning areas, (3) developing local culture curriculum that are rich in humanities, and (4) developing local culture curriculum based on parent–teacher cooperation to overcome teacher shortages at preschools. The results of this study may serve as a reference in the development of a local culture curriculum and other strategies to cultivate young children’s cultural identities.

## 1. Introduction

The rapid development of information technology in the past two decades has accelerated the pace of globalization. Encompassing a wide range of distinct political, economic, and cultural trends, the term globalization has remained at the center of contemporary political and academic discourse. Since the proliferation of the discourse surrounding globalization in the second half of the twentieth century, numerous definitions and interpretations of the term have been proposed. Some scholars view globalization as a process, whereas others view it as an outcome. Some people identify globalization with modernization or associate it with new opportunities, whereas others consider globalization to be an extension of Westernization and cultural imperialism. Generally, globalization can be defined by the growing interdependence of the world’s economies, cultures, and populations due to cross-border trade in goods and services and the increasingly dense flows of investment, people, and information [1,2,3].

Globalization has been identified as a cause of global homogenization and standardization, but globalizing forces are filtered through local contexts, thus contributing to the evolution, rather than the erasure, of local cultures. One of the goals of Taiwan’s Early Childhood Education and Care Curriculum Framework is “developing young children’s cultural identities”, which calls for attention to the development of a diverse conception of Taiwanese society and the strengthening of children’s knowledge of local cultures in Taiwan. In recent years, awareness of local culture, which is based on cultural transmission with respect to language, history, geography, diet, skills, knowledge, customs, art, and literary achievements and an appreciation of the value of local identity and traditional culture, has become a priority [4,5,6].Classes on local culture have become a crucial part of early childhood education, and they help young children better appreciate the culture styles behind their everyday lives.

Taiwan’s Early Childhood Education and Care Curriculum Framework is based on the concepts of interactions between individuals and the living environment, the shaping of children’s minds, the holistic development of children rooted in the value system of the cultural environment in which the children live, and the planning of each child’s learning process. A curriculum must be designed with consideration given to the developmental needs of children and must incorporate social and cultural activities. In Taiwan, local culture curriculum should include local cultural materials and must account for the subjective experiences of children engaging in cultural activities. Such curriculum should help children become more familiar with and appreciate the environment in which they are growing up and should cultivate their understanding and respect for other cultures [4,7,8].

The integration and implementation of local culture curriculum in Taiwanese preschools has emerged as a key focus in early childhood education in Taiwan. Accordingly, in the present study, the researcher used qualitative research methods and adopted a case study design to explore the connotations of local culture curriculum in a preschool. Focusing on the development and implementation of a vegetable-market-themed local culture curriculum at the case preschool, the researcher investigated how the teachers at the preschool combined the vision of the preschool, educational goals, and community resources to develop and implement the aforementioned curriculum as well as what problems they encountered in the process. Thereafter, the researcher developed strategies that can be used to solve such problems. The results of this study may serve as a reference in the development of local culture curriculum and the design of culturally responsive education strategies that can be used to cultivate young children’s cultural identities.

## 2. Relevant Topics

### 2.1. The Meaning of Culture

Culture has a variety of definitions; it has been defined as the spiritual creation of human beings, the material civilization created by human beings, or some combination of the two. Culture refers to a group of people’s way of life, including their religion, food, clothing, language, rites of passage, and music. It has also been defined as a system of ideas, values, beliefs, knowledge, and customs shared by a group and passed on over generations within that group in the international context. Local culture curriculum in preschools are designed to share culturally specific concepts, values, beliefs, knowledge, and customs with young children to ensure the aforementioned intergenerational transmission of culture in Taiwan [4,9,10,11,12].

### 2.2. The Relationship between Education and Culture

The cultivation of the self is always an issue of ultimate concern in the East Asian educational tradition, and education plays an important role in this process [8]. Some view education as a means of preserving, transmitting, and promoting culture [10,13]. Because of education, people benefit from the experiences of previous generations through a curriculum and pass on the knowledge they acquire to the next generation. Others view education itself as a part of culture, specifically high culture. Other scholars, especially those who hold the view that culture itself is education, argue that education is a necessary condition for culture. Indeed, culture encodes the institutions that enable human life, and education in turn ensures the preservation of such culture. Education is also necessary for one to participate in civilization [11,14]. For example, cultural preservation is a key component of education in Indigenous communities, and education is the most important channel through which Indigenous youth construct their cultural identities. Therefore, educators must strive to develop educational models suitable for Indigenous tribes or ethnic groups to ensure that Indigenous youth can effectively construct their cultural identities and inherit the culture of their predecessors [14]. In addition, Liu, Tasi, Ou and Huang (2011) studied the influence of the media on the development of gender identities among students. In the course “Gender and Media Culture”, students learned to identify the correctness of gender-related content presented in the media and to use multimedia educational materials to understand their life experiences and increase their awareness of gender and widen their range of perspectives on gender [15].These examples demonstrate that education is also necessary for an individual’s ability to participate in society [16].

### 2.3. The Meaning of Localization

Aspects of glocalization are clearly reflected in many domains of daily life. However, localization, which usually connotes a stable and highly homogenous, integrated, lasting, and unique cultural identity, is often considered a relative of globalization. Halbwachs placed localization at the intersection of geographic space and culture, arguing that local is a term used to describe relatively small areas in which people can get to know each other as well as the people in such areas themselves, who often share strong feelings and profound experiences rooted in religion, ritual, and collective memory and identity. Therefore, the concept of localization originated from the preservation of figurative geographic space and extended to abstract historical and cultural inheritance and encompasses not only the reconstruction of local cultures but also the maintenance of local identities and sense of belonging [9,17,18]. Preschool local culture curriculum with a focus on local culture-centered topics, such as Taiwan’s folklore, arts, ethnic groups, politics, economics, transportation, and ecology, can guide young children to understand their own land and culture and to appreciate and respect other local cultures, thus cultivating the children’s cultural identities [4].

### 2.4. The Meaning of Cultural Identity

Cultural identity refers to a person’s sense of belonging to a particular culture or group. The process of constructing a cultural identity involves learning about and accepting the traditions, heritage, language, religion, ancestry, aesthetics, thinking patterns, and social structures of a culture. Typically, people internalize the beliefs, values, norms, and social practices of their culture and identify themselves with that culture. Their culture becomes part of their self-conception [4,9,19].

### 2.5. Cultivating Young Children’s Cultural Identities, One of the Nine Goals in the Curriculum Outline for Preschool Education in Taiwan

Taiwan’s Ministry of Education acknowledges that early childhood education is the foundation of all later stages of education; accordingly, the Ministry introduced the Early Childhood Education and Care Curriculum Framework, which comprises a set of guidelines for preschool curriculum in Taiwan and establishes goals for the education of young children, such as supporting their physical and mental health, helping them develop good habits, enriching their life experience, promoting ethical behavior, cultivating good habits and social skills, expanding their aesthetic experiences, developing their creative thinking skills, helping them construct early childhood cultural identities, and inspiring them to care for the environment. In Taiwan, the cultivation of young children’s cultural identities is one of the nine goals of the curriculum guidelines for preschool education [4,5].

### 2.6. Factors to Consider in Developing Local Culture Curriculum in Preschools

#### 2.6.1. Preschool Location

The geographical locations of preschools, and, in turn, the living environments of the young children attending those school, vary widely. The life experienced of young children are also affected by the environments in which they live. Incorporating localized elements into the curriculum of a preschool requires an understanding of the culture and environment of the local community in which the preschool is situated. For example, compared with preschools in rural areas, preschools in urban areas have more favorable geographical locations and often have more learning resources, which facilitates the development of local culture courses [20,21].

#### 2.6.2. Preschool Curriculum Planning

The basic principles of implementing a local culture curriculum must be based on the development, needs, and interests of young children. When planning curriculum activities, educators should target the establishment of an integrated curriculum and must consider how such activities should be implemented and how relevant lessons should be taught to enhance young children’s understanding of the connotations of local culture [4,21,22].

Local culture curriculum planning should be centered on the child, and the design of educational spaces, the flexibility of teaching time, and the use of teaching materials should account for the individual abilities of young children and teachers’ teaching arrangements as well as environmental arrangements (including the space in which lessons and activities are to take place). In response to rapid globalization, the focus of curriculum planning has shifted from teachers and teaching materials to teacher guidance and adaptable lesson plans, which are more conducive to helping young children observe and learn about local cultures through firsthand experiences and develop their cultural identities [4,21,23].

#### 2.6.3. The Applications of Local Cultural Resources in Preschools

The effective use of local cultural resources can enrich a preschool curriculum. Through exploration, interaction, and learning, young children’s cultural horizons can be broadened. For example, interacting with older adults can help improve mutual intergenerational understanding and communication, and participating in festivals and reading children’s books from different regions or countries can help young children establish a mutual connection with the living environment, thereby cultivating their cultural identities [4,5,21].

#### 2.6.4. Teacher Responsibilities

Teachers play a critical role in this educational process. Preschool teachers must understand what people and things are meaningful to young children, create meaningful learning environments and activities for young children, and integrate local cultural resources into lessons and activities related to specific themes to promote young children’s learning. In addition, preschool teachers should observe young children discussing and engaging in their interests in daily life to understand what topics are of greatest interest to young children and select themes accordingly and should actively participate in teachers’ workshops or reading clubs and read widely to enhance their professional knowledge in developing a local culture curriculum. Furthermore, teachers must be sensitive to and respond to gender stereotypes implicit in life experiences, cultural practices, and media to avoid reinforcing such stereotypes when educating young children, especially when developing a local culture curriculum [4,21,24,25,26].

#### 2.6.5. The Influence of Local Culture Curriculum on Young Children’s Learning

The interactions, learnings, and life experiences of young children are mostly based on the children’s families and communities. Therefore, teaching methods that expand upon young children’s experiences by using familiar concepts and imagery may be the most effective. Materials used in local culture courses can be drawn from nearby local attractions and cultural resources with which children are familiar to draw on the children’s previous experiences and provide the children with opportunities to explore, manipulate, and repeatedly observe, thus piquing their interest in local cultures. It may help children develop a profound sense of local identity and cultural value, thus cultivating children’s cultural identities [4,21,27].

#### 2.6.6. Studies on Local Culture Curriculum in Early Childhood Education

The National Digital Library of Theses and Dissertations and the NCL (National Central Library) Taiwan Periodical Literature database were used to analyze dissertations and journal articles on local culture curriculum. A list and summary of studies on local culture curriculum in early childhood education is presented in Table 1.

These studies have provided evidence that local culture curriculum can help young children identify with and care for their own cultures and places of residence, thus cultivating their cultural identities.

## 3. Theoretical Framework

The aspects of the vegetable market–themed local culture curriculum developed and implemented at the case preschool aligned with the pedagogical insights that have been reported in the literature. The preschool principal, preschool teachers, and young children’s parents drew on local community resources to develop a vegetable market–themed local culture curriculum based on the concepts outlined in Taiwan’s Early Childhood Education and Care Curriculum Framework to cultivate the young children’s cultural identities. The framework employed in this study is illustrated in Figure 1.

## 4. Research Method

### 4.1. Qualitative Research

Qualitative research involves collecting data by using inductive methods, analyzing and interpreting the data, and understanding the perspectives of participants. Qualitative data collection methods include interviews, observations, interactions, and retrospective reviews of historical documents [36,37]. In the present study, the researcher used qualitative research methods to explore how the teachers at the case preschool integrated the vision of their preschool, educational goals, and community resources to develop and implement a local culture curriculum as well as what problems they encountered in the process. We then developed strategies to solve these problems.

### 4.2. Qualitative Case Studies

The qualitative case study method enables researchers to investigate a complex phenomenon by identifying the relevant factors and observing how they interact with each other. That is, it enables researchers to answer “how” and “why” questions with consideration of how a phenomenon is influenced by the context in which it is situated. The case study method is widely used in the field of education and the social sciences and may involve various means of data collection, including observation; interviews; surveys; document analysis; and in-depth descriptive study of an individual, family, school, group, community, region, or country [38,39,40].

In the present study, the researcher collected data through semistructured interviews conducted using an outline developed according to our research purposes. In the interviews, the respondents shared their feelings and views on the development of the local culture curriculum of the preschool at which they were employed.

### 4.3. Preschool of Case Study

The case preschool, located in Hsinchu City, Taiwan, was established in 2006, and the educational vision of the preschool is based on local culture, life experiences, and aesthetic education. Hsinchu City is a city with many distinct local cultural features, including the Hsinchu Fruit and Vegetable Market, which local residents have nicknamed “Da Cai Market.” The market is usually bustling in the early mornings and is the largest fruit and vegetable distribution center in Hsinchu City, serving as a source of products ranging from fresh vegetables, fruits, seafood, and meat products to dry goods and processed foods for stores and restaurants. Because of the major role of the market in the local culture of Hsinchu, the teachers at the case preschool elected to incorporate a visit to the market into their local culture curriculum, thus enabling their young students to feel the human touch of Hsinchu’s local culture.

The preschool is adjacent to the Hsinchu Fruit and Vegetable Market, the largest fruit and vegetable wholesale market in the area. Because the market was part of the children’s life experiences, the preschool teachers developed a vegetable-market-themed local culture curriculum.

### 4.4. Data Collection

The data sources were the interview transcripts, and the data were coded using categories we developed. We visited the case preschool to conduct semistructured interviews with the preschool’s principal and two preschool teachers on 16 June 2022. All the interviewees had been involved in the planning and design of an intergenerational learning course; during the interviews, the interviewees freely expressed their opinions regarding the course and the difficulties they had encountered in the process of its planning and implementation. The interviewees were informed of the objectives of this study before they agreed to participate. Letters of consent and interview outlines were sent to the interviewees by email. The interviews were recorded and transcribed; thereafter, the interviewees were asked to review the transcripts before the transcripts were used in this study [39,41].

Interviewees, to facilitate data collection and analysis, we recorded the interviews with the consent of the interviewees. The duration of each interview was 2 h. Table 2 and Table 3 present the demographic information of the interviewees; Table 4 and Table 5 present the interview coding method.

Each code corresponds to one interviewee and the date of their interview. For example, “Coordinator interview, A20220616” corresponds to the interview with the preschool principal conducted on 16 June 2022, whereas “Practitioner interview, A20220616” corresponds to the interview with one of the preschool teachers conducted on 16 June 2022.

The interview outline was as follows:(1)What was the underlying concept of the preschool’s vegetable-market-themed local culture curriculum?(2)What is the underlying motivation behind the preschool’s vegetable-market-themed local culture curriculum?(3)What was the process of developing the preschool’s vegetable-market-themed local culture curriculum?(4)What strategies did you use to overcome the difficulties you encountered when developing the preschool’s vegetable-market-themed local culture curriculum?(5)What changes in the young children and preschool teachers did you observe during the implementation of the vegetable-market-themed local culture curriculum?

### 4.5. Research Ethics

Before this study was conducted, a preschool principal and two preschool teachers were invited to participate in the research, which included interviews; they provided consent for the researcher to photograph the research process and to use the photos for nonprofit academic research or publication purposes. They also consented to the audio recording of their interviews and the transcription of the recordings.

## 5. The Vegetable-Market-Themed Local Culture Curriculum

### 5.1. Its Ability to Pique the Children’s Interest in History and the Humanities through the Lens of the Children’s Experiences at the Market

The principal of the preschool said, “The vegetable market–themed local culture course is integrated with human connection and history; the human connection is what the teachers look forward to the most. When we took the young children to the vegetable market, we witnessed interactions between the vegetable market owner, the proprietress, and the children. It was full of human connection, and it can even arouse different lives”. In addition, the activities and the course helped young children understand that the vegetables people eat every day are grown by farmers and enabled them to experience the hard work of farmers (coordinator interview, A20220616).

Preschool teacher A said, “ Taiwanese vegetable markets reflect specific aspects of local culture. At a vegetable market, you can witness various situations in people’s lives. There are also local delicacies and local cultural buildings at the vegetable market. People can understand the local culture by visiting the vegetable market” (practitioner interview, A20220616).

### 5.2. Its Integration of Local Culture and the Community, Specifically through Interactions between Young Children and Adults at a Vegetable Market

In response to our question regarding the underlying concept of the vegetable-market-themed local culture curriculum, the principal said, “Before the start of the vegetable-market-themed course, the teachers led planting-related activities centered on the theme of ‘Let us go: little seeds go on a trip’. At that time, the teacher’s idea was to connect the theme of planting with the vegetable-market-themed course so that the young children could understand that the vegetables planted by farmers are sold at the vegetable market. Because the largest local fruit and vegetable market in Hsinchu is located in the preschool’s community, the teachers decided to enter the community and wanted to develop a curriculum that combines local culture and community. So, they decided on the vegetable market theme” (coordinator interview, A20220616).

Preschool teacher A said, “The vegetable market curriculum provides more opportunities for children and adults to interact. It lets children see interactions in everyday life, which is meaningful” (practitioner interview, A 20220616).

## 6. Implementation of the Local Culture Curriculum

The question of how children’s familiarity with local cultures can be developed in preschool has emerged as a key topic in early childhood education in Taiwan in recent years. In addition to government authorities actively promoting education on local cultures, preschools are actively developing and implementing local culture courses for young children. In addition to enriching young children’s lives and cultural experiences, such courses can also cultivate young children’s cultural identities, which is one of the nine goals in the official curricular guidelines for preschool education in Taiwan.

Taiwan’s early childhood education philosophy is based on Confucius’s teaching of benevolence (仁). This concept is reflected throughout the Early Childhood Education and Care Curriculum Framework, a set of guidelines for preschool curricula in Taiwan. Per the framework, young children should be taught to love themselves, others, and their culture [4,5]. In the present study, we adopted a sociocultural perspective on the implementation of a local cultural curriculum at the case preschool. The approach of the case preschool serves as an example of designing local culture curriculum to cultivate young children’s cultural identities. We used qualitative research methods and adopted a case study design to explore the connotations of local culture curriculum in Taiwanese preschools.

The local culture curriculum was implemented in the following steps: (1) designing the thematic network of the vegetable-market-themed local culture course; (2) exploring young children’s previous learning experiences and adjusting the curriculum accordingly; (3) visiting and engaging in work at the vegetable market; (4) creating thank you cards and store renovation plans.

### 6.1. Design of the Theme Network of the Vegetable-Market-Themed Local-Culture Course

The principal of the preschool said, “The best way to integrate into the local area is to visit traditional vegetable markets, where you can best experience the life and culture of a region. It is a cultural trip” (coordinator interview, A 20220616). Traditional vegetable markets exist all over Taiwan and are integrated into the daily lives and culture of the people. Traditional vegetable markets may even be considered to be more representative of Taiwanese culture than night markets [42]. Therefore, the case preschool developed a local culture curriculum based on a vegetable market.

Preschool teacher A said, “The factors to be considered when designing the theme network of the vegetable-market-themed course include the children’s learning interests, the children’s individual abilities, and the teachers’ teaching beliefs. If class activities are planned in advance, preschool teachers should also pay attention to monitor the interest and abilities of young students at all times during the teaching process” (practitioner interview, A 20220616).

The main themes of the course were as follows: (1) the presentation of food on the table; (2) becoming familiar with the traditional fruit and vegetable market; (3) being grateful and polite; (4) differences between traditional and modern markets; (5) learning about vegetables, fish, and fruits; (6) vegetable and fruit palette; (7) visiting the vegetable market.

### 6.2. Exploring Young Children’s Previous Learning Experiences and Adjusting the Curriculum Accordingly

Preschool teacher B said, “At first, the theme of the TV show ‘I Stayed at the Market for a Whole Day’ piqued the children’s learning interest and let them know more about various aspects of the market, including the opening time of the market, the living environment around the market, the types of shops at the market, the price of buying and selling goods at the market, how to sell goods, and so on. When preschool teachers identify a gap between young children’s abilities and their own teaching goals, they must correct and adjust the teaching content in a timely manner, allowing the theme to continue to develop” (practitioner interview, B20220616).

### 6.3. Visiting and Engaging in Work at the Vegetable Market

The preschool teachers brought the young children to visit stores at the vegetable market, including fruit shops, grocery stores, 7–11 convenience stores, and vegetable vendors. After bringing the young children to visit the vegetable market, Preschool teacher A said, “We guided the young children to assist with the store’s sales, replenishment, and other small jobs. By experiencing the real market situation, the young children could learn through their own observation, imitation, experiences, and feelings, which helped them understand the real operation of the market and learn how to interact with others” (practitioner interview, A20220616).

Preschool teacher B said, “Visiting the vegetable market is good for young children, and young children know how to prepare before going shopping at the market.” (Practitioner interview, B20220616.)

### 6.4. Assessment Activities after Course Implementation

Preschool teacher A said, “Visiting the vegetable market is a wonderful experience and allows young children to have a practical shopping experience. Young children can experience new types of social interactions.” (Practitioner interview, A20220616.)

Preschool teacher B said, “The teacher gave the young children money to buy food at the vegetable market and told the children that the food they bought would be their own lunch, which further piqued the children’s interest in the task. Such activities can be used to evaluate whether young children have achieved their learning goals in terms of skills, cognition, and expression” (practitioner interview, B20220616).

### 6.5. Thank You Cards and Store Renovation Plans

The principal of the preschool said, “The preschool attaches great importance to interactions and etiquette among people. After the course activities, the teacher led the young children in creating thank you cards and store renovation plans. The final renovation of the store was particularly meaningful” (coordinator interview, A 20220616). Preschool teacher A said, “The young children drew a store’s products or a product location map, and the store owner posted their drawings in the store. This not only enhanced the children’s sense of achievement but also made the vegetable market more lively and helped the children give back to the store that helped them learn”. (Practitioner interview, A 20220616.)

## 7. Problems Encountered during the Development and Implementation of the Vegetable-Market-Themed Local Culture Curriculum

### 7.1. Shortages of Teachers Willing to Lead Outdoor Courses

The principal of the preschool said, “Because some local culture activities involve taking young children to visit noisy places such as vegetable markets, there needs to be a sufficient teacher–student ratio. If the teacher–student ratio was insufficient, the teachers responded by inviting the parents of the children to participate in the activity, and several teachers helped bring the young children to visit the vegetable market together”. (Coordinator interview, A 20220616.)

According to the interviews conducted in the present study, in local culture courses, the parents of young children serve as educational resources. The participation of parents can help a course run more smoothly. In addition, preschool teacher B said, “The class introduced new activities the children can do with their families, such as going to the vegetable market together over holidays. Young children can share the similarities and differences between the market they visited as part of the course and other markets. Through classroom experience and interactions with their parents, young children can improve their expression and memory”. (Practitioner interview, B 20220616.)

### 7.2. Concerns about the Safety of Young Children

Preschool teacher A said, “Because the market is only open in the morning and the shops are closed in the afternoon, young children can only visit in the morning. The market will be very crowded in the morning, and the environment of the vegetable market is noisy and slippery, which raises a lot of concerns regarding the safety of young children. If a teacher thinks that the market is not suitable for young children, after discussion, they should try to leave the market and reconsider what to do outside the market. (Practitioner interview, A 20220616.)

The principal of the preschool said, “The shops around the vegetable market are the main places to visit. Teachers should avoid going to the vegetable market at times when it will be crowded and choose the shops to visit while they are outside the vegetable market. Before the vegetable-market-themed course starts, each teacher must visit the market to survey the shops and determine the number of teachers or parents required at each location”. (Coordinator interview, A20220616.)

### 7.3. Young Children’s Lack of Experience

Preschool teacher A said, “It is the teacher’s responsibility to help children learn safely. Teachers should take good care of their students.” (Practitioner interview, A20220616.)

Preschool teacher B said, “Because the children are still young, they are inexperienced with their surroundings, especially children who rarely go out. The market is not familiar to children of this age, so we preschool teachers let the children watch a public TV program ‘I Stayed at the Market for a Whole Day,’ a documentary, which enhanced the young children’s understanding of wet markets”. (Practitioner interview, B20220616.)

## 8. Reflection, Conclusions, and Recommendations

### 8.1. Reflection

The principal of the preschool said, “In addition to the principal’s interest in choosing a theme, the opinions of each teacher are important, so the principal will invite teachers to experience the fun of the theme together. When the teachers’ interest is piqued, the curriculum will be richer. Teachers must also understand the local culture and integrate local cultural resources into the curriculum design, so we incorporated the largest fruit and vegetable market in Hsinchu, which is next to the preschool, into our curriculum” (Coordinator interview, A20220616). In addition, the principal reported witnessing changes in young children and preschool teachers during the implementation of the vegetable-market-themed local culture curriculum”, saying that “Expertise in cultural curricula has grown among teachers. The four visions of the case preschool are teaching children to love nature and the aesthetics of nature, cultivate their physical and mental health and cherish life, act with kindness and gratitude and cherish their blessings, and appreciate culture and cultural identities. Young children cherish hard-earned meals, and they also learn to express affection, enthusiasm, and courage and to have good etiquette. The children were full of sincerity, and the shopkeepers also gave full responses, from which we felt the human touch of the local culture” (Coordinator interview, A20220616).

The case preschool eliminated the restrictions on preschool textbooks from publishers and regained the ability to compile local culture curriculum textbooks. Teachers can help cultivate young children’s cultural identities through the design of local-culture curriculum to achieve one of the goals of Taiwan’s Early Childhood Education and Care Curriculum Framework. The interactions between the young children and vendors during the implementation of the vegetable-market-themed local culture curriculum also contributed to the children’s educational experience.

In the final stage of the present study, the researcher visited multiple preschools several times and observed that some preschools were biased toward science courses, such as lessons focused on cars and paper, and did not implement local culture curriculum, resulting in the students lacking awareness of and appreciation for local culture. Preschools should expand upon the conventional model of subject-oriented teaching and integrate into local communities [21]. Integrating life experiences, human history, and architecture and environments to develop children’s sense of community as well as appreciation and understanding of the characteristics and connotations of their own local culture as well as other cultures can help enhance children’s cultural awareness and identification with their own local culture [4].

### 8.2. Conclusions

The question of how we can better develop children’s familiarity with local cultures in preschools has recently emerged as a key topic in early childhood education in Taiwan. In addition to educational administrative units actively promoting education on local cultures, preschools are actively developing and implementing local culture courses for young children. In addition to enriching young children’s lives and cultural experiences, such courses can also cultivate young children’s cultural identities, which is one of the nine goals of Taiwan’s Early Childhood Education and Care Curriculum Framework. In the present study, we adopted a case study design to explore the development of a vegetable-market-themed local culture curriculum at the case preschool and investigated how the preschool teachers integrated local culture into the preschool’s vision and educational goals as well as the implementation of local culture curriculum and the obstacles the teachers encountered therein. On the basis of our conclusions, we developed recommendations for overcoming such problems and designing culturally responsive education strategies to cultivate young children’s cultural identities.

The researcher discovered that the most important aspects of the local culture curriculum were (1) its integration of local culture and community, specifically through interactions between young children and adults at a vegetable market, and (2) its ability to pique the children’s interest in history and the humanities through the lens of the children’s experiences at the market. The implementation of the local culture curriculum involved the following steps: (1) establishing a theme network; (2) planning the activities, including exploring the children’s previous experiences and adjusting the curriculum and teaching methods accordingly; (3) implementing the activities, including visiting and participating in work at the market; (4) evaluating the activities; and (5) creating thank you cards and formulating store renovation plans. Some of obstacles the teachers encountered in the development and implementation of the vegetable-market-themed local culture curriculum were (1) shortages of teachers willing to lead outdoor courses, (2) concerns about children’s safety, and (3) the young children’s lack of experience.

### 8.3. Recommendations for Designing Culturally Responsive and Inclusive Education Strategies

One of the primary challenges facing designers today is how to design curricular innovations that are appealing and useful to teachers and at the same time bring about transformative practices [43]. On the basis of our conclusions, the researcher presents the following recommendations for designing culturally responsive education strategies that can be incorporated into local culture curriculum to cultivate young children’s cultural identities.

#### 8.3.1. Integrating Local Culture and the Community

The development of a community seems simple, but it is not. Because the needs and ideas of different units in a community are diverse, conflicts of interest may arise, and establishing effective communication channels and a consensus may take time. Therefore, efforts to integrate individuals into the community should start in preschool. Developing and implementing a preschool curriculum rooted in local culture and environmental characteristics and inviting parents and community units to participate in the teaching process can help enhance students’ sense of community, and witnessing the students’ high-quality learning progress due to the community’s joint efforts can help foster trust within the community and make the community more willing to take initiative and invest resources in preschools and community development projects. The relationships between preschools and the communities that they serve are often close, especially because a preschool curriculum must focus on children and be based on the life experiences of children, and the community is where most of children’s lives and learning take place. By drawing on resources and support from the local culture and community, preschool educators must develop curricula that integrate the local culture and community to the educational benefit of the children they teach [4].

#### 8.3.2. Combining Local Culture Courses and Learning Areas

The curriculum model is based on learning areas, or areas of the classroom in which young children are permitted to explore and learn freely. (Practitioner interview, A20220616.)

When teaching about a particular theme, preschool teachers can take the opportunity to teach children about the local culture in a learning area, which allows the children to experience and explore the theme firsthand. As the principal of the case preschool said, “ The design of learning areas should start with the familiar living environments of young children, integrating relevant local cultural resources from and characteristics of the children’s families and preschools to enhance the children’s understanding and immersion in their local culture.” (Coordinator interview, A20220616.)

#### 8.3.3. Developing a Local Culture Curriculum with a Human Touch

Preschool education in general and preschool local culture courses in particular should have a human touch. As the principal of the case preschool said, “Human relationships are what preschool teachers look forward to most… (as exemplified in) the interactions between the owner of the vegetable market and the young children.” (Coordinator interview, A20220616.)

#### 8.3.4. Developing a Local Culture Curriculum Based on Parent–Teacher Cooperation to Overcome Shortages of Preschool Teachers

Strengthening the partnership between preschools, parents, and society is crucial to improving the quality of early childhood education. Parent–teacher communication can serve as a cornerstone of the implementation of local culture curriculum in preschools. In local culture courses at preschools, young children’s parents serve as valuable educational resources. Parent participation can help preschools overcome teacher shortages and help courses run more smoothly. Therefore, preschools facing teacher shortages should strive to develop local culture courses based on parent–teacher cooperation.

## Figures and Tables

**Figure 1 children-09-01789-f001:**
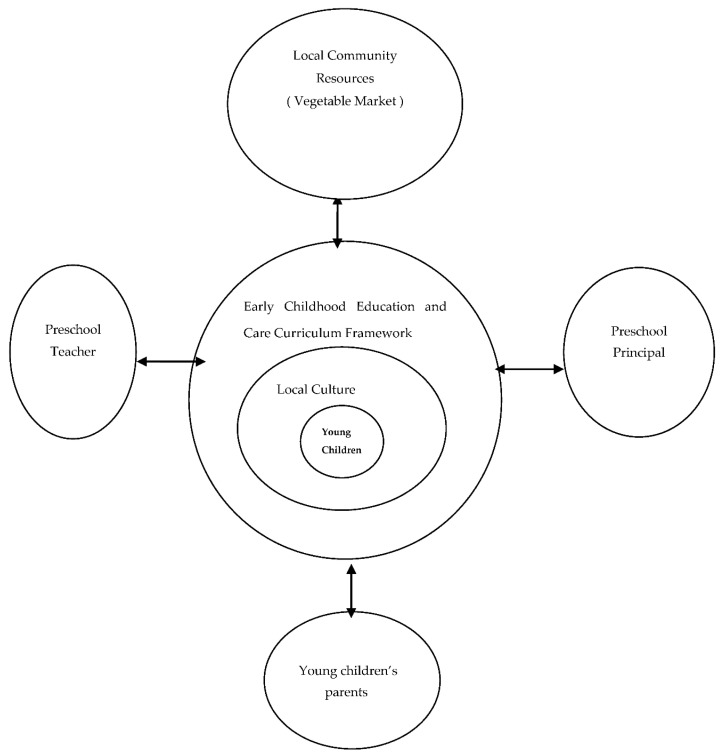
Theoretical Framework. (Source: Developed in this study).

**Table 1 children-09-01789-t001:** Studies on local culture curriculum in early childhood education.

Author	Year	Paper Title	Research Method	Study Content
Lo, Y.F. [25]	2013	Action Research on Meaningful Learning of Localized Curriculum in a Preschool	Action research	This study explored how a local culture curriculum can cultivate young children’s cultural identities. A preschool teacher arranged for their young students to participate in a cultural celebration off campus and create small books based on their experiences.
Wang, Y.T. [28]	2013	A Study of Teacher’s Decision-Making Process and Profession Growth in the Implementation of Localized Curriculum in Preschool	Case study	After implementing a localized curriculum, teachers assessed their ownprofessional growth in the areas of curriculum planning and evaluation, communicative and collaborative performance during team teaching, and reflective thinking.
Tseng, L.T. [29]	2015	Integrating Drama Teaching into Localized Curriculum: A Case Study	Case study	This study demonstrated that drama activities can cultivate children’s cultural identities and improve their expression and thinking skills. Through drama teaching, teachers and coteachers can enhance their ability to pose guiding questions, lead discussions, and integrate drama into theme-based instruction.
Lai, H.Y. [30]	2015	The Study of Using Community Resources to Implement Localized Curriculum in Preschool	Action research	This study demonstrated that a localized preschool curriculum can enhance young children’s cultural identities and connection to theircommunities.
Lin, S.C., and Guo, L.P. [31]	2016	The Teaching Course of the Curriculum for Preschool Based on the Educare Activity Curriculum Outline—to the Local Courses as Example	Action research	Themes of local culture courses are often centered on young children’sinterests and life experiences. Combining curricular activities and community life can help young children achieve a sense of spontaneityand happiness.
Chen, S.Y. [32]	2016	An Exploration of Implementing Community Resources on Thematic Curriculum	Case study	Preschool teachers should learn more about community resources, organize community resource networks, and continue to interact with the community or engage in dialogues with local experts. Providing parents with the opportunity to participate in the teaching process can help preschools facilitates the use of community resources in teaching and promotes parent–child interaction.
Hu, M.C., and Cheng, Y.L. [27]	2017	The Process of Implementing Local Cultural Curriculum with Culturally Responsive Practice in Learning Areas of a Truku Preschool	Case study	The findings of this study were as follows: (a) the preschool educatorsimplemented culturally responsive curriculum practices by establishing culturally relevant themes, placing culturally relevant materials in learning areas, and incorporating teacher-made materials to facilitate cultural responsiveness; (b) by adopting a local-culture-integrated curriculum, the educators could nurture the problem-solving abilities of young Truku children; and (c) when they encountered difficulties, including the lack of storybooks and teaching materials related to Truku culture in the community, the educators attempted to produce culturally relevant teaching materials or picture books by themselves.
Wu, L.M. [33]	2018	The Predicament and Prospect of Implementing the Curriculum of the Culture in the Preschool	Case study	In this study, teachers implemented an integrated preschool curriculum based on localized teaching materials from the students’ own culture.
Hsu, C.J. [34]	2019	Action Research on Implementing Localized Cultural Curriculum in the Preschool	Action research	Through local-culture courses, young children can learn to identify with and care for the place where they live.
Lin, Y.Y. [20]	2019	Localization Courses Start from the Preschool	Action research	In this study, preschool teachers started at the location of the preschool and guided their young students in exploring their everyday lives, drawing on the students’ previous experiences and incorporating emotional connections into the course theme. The teachers effectively used local cultural resources and portrayed all people and things as nutrients that help individuals grow.
Liu, S.F. [35]	2021	A Study of Integration and Utilization of Community Resources in a Localized Curriculum: The Case of a Nonprofit Preschool in Taoyuan City	Action research	In this study, teachers collected various community resources and incorporated them into courses based on the experiences of young children.

(Source: Chen, 2016 [32]; Hsu, 2019 [34]; Hsu & Cheng, 2017 [27]; Lai, 2015 [30]; Lin, 2019 [20]; Lin & Guo, 2016 [31]; Liu, 2021 [35]; Lo, 2013 [28]; Shih & Wang, 2022 [21]; Tseng, 2015 [29]; Wang, 2013 [28]; Wu, 2018 [33]).

**Table 2 children-09-01789-t002:** Coordinator of local culture curriculum.

Job Title	Sex	Age	Education Level	Work Experience
Preschool principal	Male	59 years	Bachelor of Early Childhood Education	Preschool education: 32 years;preschool principal: 16 years.

(Source: the researcher).

**Table 3 children-09-01789-t003:** Practitioners of local culture curriculum.

Job Title	Sex	Age	Education Level	Work Experience
Preschool teacher	Female	42 years	Bachelor of Early Childhood Education	Early childhood education: 16 years
Preschool teacher	Female	39 years	Bachelor of Early Childhood Education	Early childhood education: 15 years

(Source: the researcher).

**Table 4 children-09-01789-t004:** Interview codes.

Interviewee		Code
Coordinator:	preschool principal	Coordinator interview, A20220616
Practitioner:	A preschool teacher	Practitioner interview, A20220616
	B preschool teacher	Practitioner interview, B20220616

(Source: the researcher).

**Table 5 children-09-01789-t005:** Interview codes.

Interviewee		Code
Coordinator:	preschool principal	Coordinator interview, A20220616
Practitioner:	A preschool teacher	Practitioner interview, A20220616
	B preschool teacher	Practitioner interview, B20220616

(Source: the researcher).

## Data Availability

Not applicable.

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
