# Peer review of "Designing Culturally Responsive Education Strategies to Cultivate Young Children’s Cultural Identities: A Case Study of the Development of a Preschool Local Culture Curriculum"

_children, 2022, doi:10.3390/children9121789_

Round 1

Reviewer 1 Report

The present form of the paper does not do justice to the concept and to the activity that the school carried out. While I encourage you to continue working in this direction you need to invest much more on the theoretical framework and reflect on what can this activity offer in terms of pedagogy and holistic education of children. I encourage you to read and reflect on the following issues before re-writing and resubmitting  to this or any other journal you need to

1. identify and clarify the pedagogical theory in which you are grounding your research - I suggest that you take a closer look at sociocultural pedagogies, in particular, situated and distributed learning and/or J. Gibson's affordances theory,

2.  you will probably need to clarify the philosophy of education adopted by the ministry documents

3. you need to clarify the specific pedagogical philosophy that the school makes use of. are there any foundational documents?

4. you need to justify and explain why a visit to the market is to be considered a cultural trip - in what way are children immersing themselves in local culture? - you take this for granted, however, the cultural element was not highlighted either the methodology section nor in the responses that your report

5. Tied to the above:  you need to clarify which pedagogies were used; which themes were explored. What is specific to the Taiwanese market? - how does it reflect local culture? In response of one of the interviewees, one may infer that the school makes use of an emergent curriculum approach.  Is this the case? 

6.  Section 2.6.6. is out for synch with the whole paper - in no way was the methodology (systematic literature review?) used to gather this data explained. Nor was the reason why this data was collected or how did it influence your reflections made clear. It has no place in the current form of the paper

7. The methodology section needs substantial revision. Which approach and theoretical framework did you use? Why did you identify this particular school? Which analytical methods did you adopt

8. Likewise the analysis needs to be clear. Answering each question that you posed is not enough you need to delve deeper and build on the literature that was explored at the beginning of the paper.

I hope the above helps you to re-write and better reflect on the experience which I believe is a valid one and which merits in-depth reflection.

Author Response

Author Response Letter

Dear Reviewer

The author of this manuscript (Manuscript ID: children-1998241) has modified this manuscript according to the reviewer’s comments.

I resubmit this manuscript.

Thank you for the reviewer’s comments.

The author’s response letter illustrates the revision of this manuscript.

Manuscript ID: children-1998241

Designing Culturally Responsive Education Strategies to Cultivate Young Children's Cultural Identities: A Case Study of the Development of a Preschool Local Culture Curriculum

The first reviewer’s comments

Modified

page

Revise

1.     identify and clarify the pedagogical theory in which you are grounding your research - I suggest that you take a closer look at sociocultural pedagogies, in particular, situated and distributed learning and/or J. Gibson's affordances theory,

   10

Thanks to the reviewer’s opinion

.

Modified according to the reviewer’s opinion. Modify the page as shown on page 10.

Line 359-374.

2.     you will probably need to clarify the philosophy of education adopted by the ministry documents

10

Thanks to the reviewer’s opinion

.

Modified according to the reviewer’s opinion. Modify the page as shown on page 10.

Line 359-363.

3.     you need to clarify the specific pedagogical philosophy that the school makes use of. are there any foundational documents?

10

Thanks to the reviewer’s opinion

.

Modified according to the reviewer’s opinion. Modify the page as shown on page 10.

Line 359-363.

4.     you need to justify and explain why a visit to the market is to be considered a cultural trip - in what way are children immersing themselves in local culture? - you take this for granted, however, the cultural element was not highlighted either the methodology section nor in the responses that your report

10

Thanks to the reviewer’s opinion

.

Modified according to the reviewer’s opinion. Modify the page as shown on page 10.

Line 374-376.

5. Tied to the above:  you need to clarify which pedagogies were used; which themes were explored. What is specific to the Taiwanese market? - how does it reflect local culture? In response of one of the interviewees, one may infer that the school makes use of an emergent curriculum approach.  Is this the case? 

9

   10

   11

   12

   14

Thanks to the reviewer’s opinion

.

Modified according to the reviewer’s opinion. Modify the page as shown on pages 9, 10, 11, 12, 14.

Line 331-335.

Line 348-350.

Line 374-376.

Line 411-412.

Line 466-467.

Line 559-560.

6.     Section 2.6.6. is out for synch with the whole paper - in no way was the methodology (systematic literature review?) used to gather this data explained. Nor was the reason why this data was collected or how did it influence your reflections made clear. It has no place in the current form of the paper

5

   6

Thanks to the reviewer’s opinion

.

Modified according to the reviewer’s opinion. Modify the page as shown on pages 5,6.

Line 219-220.

Line 226-228.

7. The methodology section needs substantial revision. Which approach and theoretical framework did you use? Why did you identify this particular school? Which analytical methods did you adopt

6

7

   8

Thanks to the reviewer’s opinion

.

Modified according to the reviewer’s opinion. Modify the page as shown on pages 6, 7, 8.

.

 Thank you for the reviewer’s comments.

Reviewer 2 Report

Overall, I think this paper will be of interest to the readers and presents interesting findings. However, please respond to these comments and make related revisions. 

Some Specific Comments

Abstract – the researcher (singular) but ‘our conclusions’

Line 55 – cultural context?

I wonder if the framing discussion of globalisation could offer more critical discussion and support from literature.

Similarly, the meaning of culture discussion lacks complexity and viewpoints.

Line 102 – give examples of this and support with a source.

Lines 127-128 – something needs fixed about the spacing

The reader can work out where the case school is based on the descriptions, as well as who the participants are (particularly the principal). How does this sit with any ethical considerations or assurances given to participants?

Lines 282-292 – I don’t think you need to have the extended heading numbers? Perhaps bullet the list?

General

This is well written, and a number of interesting findings emerge about the benefits and issues relating to this approach to learning. The recommendations emerge clearly from the data.

Make it clearer to the reader when discussion pertains to local or international contexts (or both). E.g lines 86-90 – is this about the Taiwanese context alone?

Instead of talking of scholars and other scholars you need to cite literature (e.g. line 95)

Literature and sources tend to appear at end of complex paragraphs. Can you weave supporting literature into the writing more equally and in a way that more clearly links literature to specific claims being made? I think policy literature and wider research into curricula development should be cited separately. This could provide more opportunity for comparison and critique.

I would provide more context – given this is based on Taiwanese education what is the cultural composition of Taiwan. I’m not sure what is meant by ‘local culture’ and if this is being contrasted with global culture (and indeed what this may be)? In what ways does the market exemplify local culture? Unpacking these questions would help the reader. How diverse is Taiwain? In the curricular context where does the culture curriculum stand alongside other disciplines and subjects? Is this a new addition to the curriculum? (can you be more specific than ‘recent years’?) Line 430 suggest an intriguing issue which needs to be explored and discussed. What were these restrictions, why and when were they imposed? This should be discussed earlier in the paper.

Author Response

Author Response Letter

Dear Reviewer

The author of this manuscript (Manuscript ID: children-1998241) has modified this manuscript according to the reviewer’s comments.

I resubmit this manuscript.

Thank you for the reviewer’s comments.

The author’s response letter illustrates the revision of this manuscript.

Manuscript ID: children-1998241

Designing Culturally Responsive Education Strategies to Cultivate Young Children's Cultural Identities: A Case Study of the Development of a Preschool Local Culture Curriculum

The second reviewer’s comments

Modified

page

Revise

Overall, I think this paper will be of interest to the readers and presents interesting findings.

This is well written, and a number of interesting findings emerge about the benefits and issues relating to this approach to learning. The recommendations emerge clearly from the data.

Thanks to the editor’s affirmation.

Abstract – the researcher (singular) but ‘our conclusions’

1

Thanks to the reviewer’s opinion

.

Modified according to the reviewer’s opinion. Modify the page as shown on page 1.

Line 24.

Line 55 – cultural context?

   2

Thanks to the reviewer’s opinion

.

Modified according to the reviewer’s opinion.

The author revise cultural styles

Modify the page as shown on page 2.

Line 58.

Line 102 – give examples of this and support with a source.

3

Thanks to the reviewer’s opinion

.

Modified according to the reviewer’s opinion. Modify the page as shown on page 3.

Line 107-119.

Lines 127-128 – something needs fixed about the spacing

Thanks to the reviewer’s opinion

.

Modified according to the reviewer’s opinion.

The reader can work out where the case school is based on the descriptions, as well as who the participants are (particularly the principal). How does this sit with any ethical considerations or assurances given to participants?

9

Thanks to the reviewer’s opinion

.

Modified according to the reviewer’s opinion. Modify the page as shown on page 9.

Line 315-319.

Lines 282-292 – I don’t think you need to have the extended heading numbers? Perhaps bullet the list?

9

Thanks to the reviewer’s opinion

.

Modified according to the reviewer’s opinion. Modify the page as shown on page 9.

Line 303-313.

Make it clearer to the reader when discussion pertains to local or international contexts (or both). E.g lines 86-90 – is this about the Taiwanese context alone?

2

Thanks to the reviewer’s opinion

.

Modified according to the reviewer’s opinion. Modify the page as shown on page 2.

Line 91.

Line 94.

Instead of talking of scholars and other scholars you need to cite literature (e.g. line 95)

3

Thanks to the reviewer’s opinion

.

Modified according to the reviewer’s opinion. Modify the page as shown on page 3.

Line 99-100.

Literature and sources tend to appear at end of complex paragraphs. Can you weave supporting literature into the writing more equally and in a way that more clearly links literature to specific claims being made? I think policy literature and wider research into curricula development should be cited separately. This could provide more opportunity forcomparison and critique.

Thanks to the reviewer’s opinion

.

Modified according to the reviewer’s opinion.

I would provide more context – given this is based on Taiwanese education what is the cultural composition of Taiwan. I’m not sure what is meant by ‘local culture’ and if this is being contrasted with global culture (and indeed what this may be)? In what ways does the market exemplify local culture? Unpacking these questions would help the reader. How diverse is Taiwain? In the curricular context where does the culture curriculum stand alongside other disciplines and subjects? Is this a new addition to the curriculum? (can you be more specific than ‘recent years’?) Line 430 suggest an intriguing issue which needs to be explored and discussed. What were these restrictions, why and when were they imposed? This should be discussed earlier in the paper.

2

7

Thanks to the reviewer’s opinion

.

Modified according to the reviewer’s opinion. Modify the page as shown on pages 2,7.

Line 91-94.

Line 257-256.

 Thank you for the reviewer’s comments.

Round 2

Reviewer 1 Report

Thank you for taking into consideration my comments. In particular, I note that you did clarify the theoretical framework adopted, the philosophy of the school, and the contribution of interviewees, however, I still believe that there is room for improvement.  Please go through my original comments, once again.  In addition to those comments I am highlighting that

I still believe that Section 2.6.6. is out of place,

I appreciate the link with the Confucian philosophy of education (which highlighted the main themes) these need to be clarified more - do not take for granted that your readers are aware of what you are taling about

I am not sure that the papers on aboriginal culture make justice to the concept that you want to promote. You are interested in culture - works on situated learning would probably be more suited. The reflections on this pedagogy should be foundational to your analysis

Why should your readers be interested in what you are doing - how can they benefit from your practice and reflection

Author Response

Author Response Letter

Dear Reviewer

The author of this manuscript (Manuscript ID: children-1998241) has modified this manuscript according to the reviewer’s comments.

I resubmit this manuscript.

Thank you for the reviewer’s comments.

The author’s response letter illustrates the revision of this manuscript.

Manuscript ID: children-1998241

Designing Culturally Responsive Education Strategies to Cultivate Young Children's Cultural Identities: A Case Study of the Development of a Preschool Local Culture Curriculum

The first reviewer’s comments

Modified

page

Revise

1.     Thank you for taking into consideration my comments. In particular, I note that you did clarify the theoretical framework adopted, the philosophy of the school, and the contribution of interviewees.

Thanks to the reviewer’s affirmation.

2.     however, I still believe that there is room for improvement.  Please go through my original comments, once again. In addition to those comments I am highlighting that. I still believe that Section 2.6.6. is out of place,

6

7

Thanks to the reviewer’s opinion

.

Modified according to the reviewer’s opinion. Modify the page as shown on pages 6,7.

Line 229-257.

3.     I appreciate the link with the Confucian philosophy of education (which highlighted the main themes) these need to be clarified more - do not take for granted that your readers are aware of what you are talking about.

11

Thanks to the reviewer’s affirmation.

Modified according to the reviewer’s opinion. Modify the page as shown on page 11.

Line 387-396.

4.     I am not sure that the papers on aboriginal culture make justice to the concept that you want to promote. You are interested in culture - works on situated learning would probably be more suited. The reflections on this pedagogy should be foundational to your analysis

6

7

Thanks to the reviewer’s opinion

.

Modified according to the reviewer’s opinion. Modify the page as shown on pages 6, 7.

Line 230-235.

Thank you for the reviewer’s comments.
